# Effects of Substituting Fishmeal with Soy Protein Concentrate Supplemented with Essential Amino Acids in the Olive Flounder (*Paralichthys olivaceus*) Diet on the Expression of Genes Related to Growth, Stress, Immunity, and Digestive Enzyme

**DOI:** 10.3390/ani14203039

**Published:** 2024-10-21

**Authors:** Sang Hyun Lee, Jong-Won Park, Minhwan Jeong, Dain Lee, Julan Kim, Hyun-Chul Kim

**Affiliations:** Genetics and Breeding Research Center, National Institute of Fisheries Science, Geoje 53334, Republic of Korea; tyier5883@naver.com (S.H.L.); dapowind@korea.kr (J.-W.P.); mhjeong5920@korea.kr (M.J.); gene419@korea.kr (D.L.); kimjulan@korea.kr (J.K.)

**Keywords:** fishmeal substitution, soy protein concentrate, *Paralichthys olivaceus*, gene expression, growth performance, feed utilization

## Abstract

This experiment was conducted in accordance with the prevailing trend of identifying alternative sources of nutrition to compensate for the economic shortcomings of fish meal in the aquaculture industry. The experiment employed soybean-based concentrated protein, which is approximately half the cost of fish meal, and focused on investigating its impact on growth and physiological processes. Three experimental diets were prepared: the control diet (Con) contained 60% FM, and the experimental diets contained 25% (SPC25) and 50% (SPC50) FM replaced by SPC. Following the 140-day feeding period, no significant differences in growth performance were observed. However, an increase in the replacement rate was associated with a reduction in stress and immune sensitivity, while the SPC25 treatment exhibited elevated levels of digestive enzyme-related genes. Therefore, substituting 25% FM with SPC was most effective for commercial fish farming.

## 1. Introduction

Olive flounder (*Paralichthys olivaceus*) is a popular and commercially important marine fish species in the aquaculture industry in Republic of Korea (henceforth, Korea) [1,2], accounting for 50% of aquaculture production in 2023 [2]. Feed is an important component in olive flounder farming because it makes up 30–35% of total operational costs [3]. In particular, protein is an important nutrient in the diets of carnivorous fish species such as olive flounder [4]. Fish meal (FM) is used as the main protein source and 40–70% protein is supplied with FM in carnivorous fish feeds [4]. FM has advantages, such as a balanced amino acid (AA) profile and high nutrient digestibility [5,6]. However, its production has been decreasing or plateauing in recent decades due to overfishing of fish stocks and its price is rising due to increased global demand [5,7]. Therefore, it is important to search for a replacement that is inexpensive and available year-round.

Soybeans are less expensive than FM but contain non-starch polysaccharides, which are difficult for fish to digest, and high levels of anti-nutritional factors such as trypsin inhibitors [8,9]. Soy protein concentrate (SPC) is obtained after soybean meal is processed and is almost free of anti-nutritional factors [10]. It also has higher nutrient digestibility than soybean meal and can be used to replace a substantial amount of FM in fish feed [11,12,13]. Nevertheless, Lim et al. [14] reported that plant protein sources are more likely to lack some essential amino acids (EAA) and fatty acids (FAs) compared to FM, which would adversely affect fish growth. Therefore, substituting FM with a plant protein source is limited to some extent in fish feeds [15,16]. Many attempts have been made to replace FM with various plant protein sources [16,17]. Deng et al. [13] reported that the growth performance of olive flounder fed a diet replacing 75% FM with SPC and providing an AA supplement was similar to fish fed a diet replacing 25% FM with SPC without AA supplementation but inferior to fish fed a 74% FM-based diet.

Growth performance and feed efficiency of red sea bream (*Pagrus major*) fed the SPC diet with AA supplementation were similar or slightly lower than the values of fish fed a diet replacing FM with SPC and providing a AA supplement [18]. Kokou et al. [19] reported that supplementing with AAs improves all growth and protein conversion ratios up to the 40 substitution levels (based on 58% FM) in the gilthead seabream (*Sparus aurata* L.).

Previous studies have reported changes in the expression of growth and immune-related genes when FM is replaced with other ingredients, such as SPC. In Moon et al. [20], the expression of growth hormone-releasing hormone and insulin-like growth factor-I (IGF-I) genes changed in fish when 30% of FM was replaced with SPC based on 60% FM. In another study, as a result of replacing 14.5% with SPC based on 65% FM, the expression levels of the immune-related genes interleukin (IL)-10 and IL-1β were higher in a low FM diet and induced a more balanced immune response when the low FM diet was consumed by olive flounder [21].

Therefore, this study investigated the effect of substituting FM with SPC in the diet along with providing AA supplementation on the expression of growth, stress, immune, and digestive enzyme-related genes in *P. olivaceus*.

## 2. Materials and Methods

### 2.1. Experimental Conditions

A total of 900 olive flounder (initial mean weight 727.8 ± 7.89 g; mean ± SE) were purchased from the Genetics and Breeding Research Center (National Fisheries Research Institute, Geoje-si, Gyeongsangnam-do, Korea) and transported to the Feed Research Center (National Fisheries Research Institute, Pohang-si, Gyeongsangbuk-do, Republic of Korea). The fish underwent a 2-week acclimation period, during which they were fed commercial feed (crude protein: 54.0% and crude lipid: 10.0%) (Suhyup Feed, Uiryeong-gun, Gyeongsangnam-do, Republic of Korea) under experimental conditions. After the 2-week acclimatization period, the fish were randomly distributed into 9 8-ton flow-through tanks (water volume: 5 tons) (100 fish/tank; *n* = 3 tanks per treatment). The water source was sand-filtered seawater. The water temperature ranged from 16.5 °C to 22.4 °C (19.9 ± 3.4 °C; mean ± SD) and dissolved oxygen (8.5 ± 1.0 mg/L; mean ± SD) was supplied directly to each tank by an oxygen generator. Dead fish were immediately removed from each tank after measuring length and weight throughout the feeding trial.

### 2.2. Experimental Diet Preparation

Three experimental isonitrogenous (53.0%) and isoenergetic (3.9 kcal/g diet) diets were formulated (Table 1) to meet the dietary protein and lipid requirements of olive flounder [22,23]. The control (Con) diet contained 60% FM. In the other diets, 25% (SPC_25_) and 50% (SPC_50_) FM were substituted with SPC supplemented with lysine (0.375 and 0.750%) and methionine (0.145 and 0.290%, respectively) (MP Bio, Irvine, CA, USA), respectively. The method for preparing and storing diet was based on synthesis of findings from previous studies [13,15].

The ingredients in the experimental diets were thoroughly blended with water at a ratio of 5:1. The pressure control of the experimental feed extruder (twin screw extruder (ATX-II, Fesco, Daegu, Korea) was controlled only by the screw speed (rpm/min), while keeping the barrel temperature (115–130 °C), conditioner temperature (80 °C), steam (31.6 kg/h), and feed rate (50 kg/h) constant (low pressure, 885 rpm/min; high pressure: high pressure, 708 rpm/min). The speed of the discs (rpm/min) and rotors (rotor, rpm/min) from the air classifier mill (XP 50, (C) KMTECH, Guri-si, Gyeonggi-do, Republic of Korea) were adjusted accordingly. All experimental diets were stored at −20 °C until use. The experimental diets were fed to the fish twice daily (08:00 and 17:00) to apparent satiation for 140 days. The nutrient requirements of the fish were satisfied by all the experimental diets [22,23].

### 2.3. Determination of Biological Indices 

Six fish from each tank were sampled at the beginning and after the 140-day experiment to measure the biological indices (CF, VSI, and HSI). All live fish were starved for 24 h and anesthetized in 50 ppm 2-phenoxyethanol (Sigma, St. Louis, MO, USA) before measurement. All live fish from each tank were counted and weighed collectively to measure survival and weight gain at the beginning and end of the experiment. The biological indices of the fish were calculated using the following equations:Specific growth rate (SGR, %/day) = (Ln final weight − Ln initial weight) × 100/day feeding trial(1)
Daily feed intake (%/day) = feed consumption × 100/(initial fish weight + final fish weight + dead fish weight) (2)
Feed efficiency (FE) = weight gain/feed consumption(3)
Protein efficiency ratio (PER) = weight gain/protein consumption (4)
Protein retention (PR, %) = protein gain × 100/protein consumption(5)
Condition factor (CF, g/cm^3^) = body weight (g) × 100/body length (cm)^3^(6)
Viscerosomatic index (VSI, %) = visceral weight × 100/body weight(7)
Hepatosomatic index (HSI, %) = liver weight × 100/body weight(8)

### 2.4. Biochemical Composition of the Experimental Diets and the Fish

The dorsal muscle of six fish at the beginning of the experiment and after completing the 140-day feeding experiment was homogenized and used to analyze the biochemical composition. The proximate composition of the experimental diets and the dorsal muscle were analyzed according to standard methods [25]. Crude protein and crude lipid content were determined using the Kjeldahl method (Kjeltec^TM^ 2100 Distillation Unit; Foss, Hillerød, Denmark) and ether-extraction (Soxtec^TM^ 2043 Fat Extraction System; Foss, Hillerød, Denmark), respectively. Moisture content was measured by drying in an oven for 24 h at 105 °C, and the ash content of the samples was determined using a muffle furnace at 550 °C for 4 h.

The AAs in the experimental feeds were hydrolyzed by adding 20 mL 6 N HCl at 110 °C for 24 h in a drying oven. The solution was filtered with a glass filter and concentrated under decompression at 55 °C to completely evaporate the acid and water. The sample was dissolved with sodium citrate buffer (pH 2.20) in a 25 mL flask, filtered with a 0.45 µm membrane filter, and analyzed using an automatic AA analyzer (Biochrom 30+; Biochrom Ltd., Cambridge, UK).

Total lipids in the experimental feeds were extracted with a chloroform and methanol mixture (2:1) to methylate FA with a 14% BF-methanol (Sigma-Aldrich, St. Louis, MO, USA) solution and FAs were analyzed by gas chromatography (Trace 1310; Thermo Scientific, Waltham, MA, USA) equipped with a capillary column (SPTM-2560, 100 m × 0.25 mm; Supelco Inc., Bellefonte, PA, USA). A mixture of 37 FAs (PUFA 37 Component FAME Mix; Supelco) was used as the FA standard. The methods and procedures for the FA analysis are described by [26].

### 2.5. Quantitative Real-Time Polymerase Chain Reaction (PCR)

The brain, stomach, and middle intestine of six randomly selected fish were collected at the end of the 140-day feeding trial, immersed in Trizol reagent (Ambion, Carlsbad, CA, USA) containing five times the reagent tissue volume, and stored at –80 °C until use. The tissue was thawed just before RNA extraction, and the extracted RNA was synthesized using RT and the GO Master Mix (MPbio, Irvine, CA, USA). The concentration of cDNA used was 100 ng/µL. The olive flounder 18S rRNA gene (Genbank, EF126037.1) was selected as the housekeeping gene and all primers used for gene expression are shown in Table 2. The levels of gene expression induced by the dietary treatments were compared among the growth, immune, and stress-related genes using RNA extracted from the brain. The amount of gene expression caused by the dietary treatments was compared through digestive enzyme genes using RNA extracted from the stomach and middle intestine. These genes underwent an initial denaturation step at 98 °C for 2 min, followed by denaturation at 98 °C for 10 s, annealing at 60 °C for 10 s, and extension at 68 °C for 30 s for 40 cycles.

### 2.6. Statistical Analysis

Statistical analysis was carried out using R program 3.5.1 (R Development Core Team, 2018, Vienna, Austria) and SPSS version 24.0 software (SPSS Inc., Chicago, IL, USA) to conduct one-way analysis of variance and the Duncan’s multiple range test [27]. All values are expressed as means of triplicates ± SEs. A *p*-value < 0.05 was considered significant.

## 3. Results

### 3.1. AA and FA Profiles of the Experimental Diets

The AA and FA profiles of the experimental diets are shown in Table 3 and Table 4, respectively. The contents of EAAs, such as lysine and methionine, were lower in the SPC than in the FM. Histidine and phenylalanine content among EAAs and glutamic acid and proline content among NEAAs increased as the amount of substituted SPC was increased, but other EAAs and NEAAs decreased. The arginine (2.04–2.10% of the diet) and lysine (1.50–2.10% of the diet) contents in all of the experimental diets met the dietary requirements of olive flounder [28,29] but methionine (1.44–1.49% of the diet) was lower than the dietary requirement [30].

FAs were not detected in SPC. The total content of saturated FAs (∑SFA) increased as the amount of SPC substituted for FM increased but decreased with the total content of monounsaturated FAs (∑MUFA) and n-3 highly unsaturated FAs (∑n-3-HUFA). The ∑n-3 HUFA content in the SPC_50_ diet did not satisfy the dietary ∑n-3 HUFA requirement of olive flounder [31].

### 3.2. Survival and Growth Performance

The survival and growth performance results are shown in Table 5. The survival rate ranged from 94.8% to 96.3% but it was not affected by dietary treatment. Weight gain ranged from 390.8 to 442.3 g/fish and the SGR was 0.31–0.34%/day but these parameters were not affected by replacing FM with SPC.

### 3.3. Feed Availability and Biological Indices

The feed availability and biological indices results are shown in Table 6. Daily feed intake (DFI) ranged from 0.44% to 0.48%/day and was not affected by dietary treatment. FER was 0.63–0.70, PER was 1.16–1.32, and PR was 37.36–42.90%. None of the feed availability indices changed in response to dietary treatment.

The condition factor of the fish ranged from 1.04 to 1.16 g/cm^3^, the VSI was 3.45–3.99%, and the HSI was 1.44–1.57%. These parameters were not affected by dietary treatment.

### 3.4. Proximate Composition of Dorsal Muscle

The proximate composition of dorsal muscle result is shown in Table 7. The moisture content ranged from 71.95% to 72.87% and crude protein was 21.91–22.44%. These did not change after substituting SPC for FM. However, lipid levels of fish fed the SPC_50_ diet were significantly (*p* < 0.04) higher than those of fish fed the Con and SPC_25_ diets. Ash content of Con and SPC_25_ fish was significantly (*p* < 0.05) higher than that of SPC_50_ fish.

### 3.5. Expression Analysis of Growth, Immune, and Stress-Related Genes

The mRNA gene expression results of growth, immune, and stress-related genes are shown in Figure 1. The expression levels of genes encoding insulin-like growth factor (IGF), growth factor beta-3-like protein (GFB-3), interleukin-8 (IL-8), and caspase in the brains of fish fed the Con diet were significantly (all *p* < 0.001) higher than those of fish fed the SPC_25_ and SPC_50_ diets.

The expression levels of genes related to stress, including superoxide dismutase (SOD), glutathione peroxidase (GPX), peroxiredoxin (PRX), thioredoxin (TRX), were significantly (*p* < 0.05) higher in Con fish than other groups. However, that of heat shock protein 70 (HSP70) was not different among the diets.

### 3.6. Expression Analysis of Digestive Enzyme Genes in the Stomach and Middle Intestine 

The mRNA gene expression results of digestive enzyme genes in stomach and middle intestine are shown in Figure 2 and Figure 3. The expression of the genes encoding α-amylase (Amy), chymotrypsinogen 2 (chymo-TRY2), trypsinogen 2 (TRY2), and trypsinogen 3 (TRY3) and lipase activity in the stomach and middle intestine of olive flounder are presented in Figure 2 and Figure 3, respectively. Amy and lipase in the stomach were not different among the experimental diets. However, the expression levels of chymo-TRY2, TRY2, and TRY3 in the stomachs of SPC_25_ fish were significantly (all *p* < 0.02 or better) higher than those of Con and SPC_50_ fish.

In the middle intestine, the expressions of Amy, chymo-TRY2, and TRY2 were significantly higher in Con, Con and SPC_25_, and Con and SPC_25_ fish, respectively, than in the remaining group(s) in each comparison. The TRY3 gene was not affected by dietary treatment. Lipase activity was significantly higher in SPC_25_ fish than in Con fish but not SPC_50_ fish.

## 4. Discussion

The weight gain and SGR of olive flounder fed the SPC_25_ and SPC_50_ diets were comparable to fish fed the Con diet, suggesting that up to 50% of the FM could be replaced by SPC with AA (lysine and methionine) supplementation in a 60% FM-based diet without affecting growth performance. This contrasts with a previous study that did not include AA supplementation and found that using FM lowered growth in this species [13]. SPC can replace approximately 40% of the FM in the yellow croaker (*Larimichthys crocea*) diet and 60% of the FM with AA supplementation can be substituted in the gilthead sea bream diet [11,19]. Similarly, 10% and 40% FM could be substituted with corn gluten meal without AA supplementation in the Asian seabass (*Lates calcarifer*) diet [32] and with AA (arginine, lysine, and tryptophan) supplementation in the olive flounder diet [33] without deteriorating growth performance.

The arginine (2.04–2.10% of the diet) [28] and lysine (1.50–2.10% of the diet) [29] requirements of olive flounder were satisfied in all experimental diets in the current study. However, methionine content (1.17–1.37% of the diet) was slightly lower than the dietary methionine requirement in all experimental diets including the Con diet (1.44–1.49% of the diet in the presence of 0.06% cysteine) [30]. However, previous studies have demonstrated that cysteine can spare about 40–50% of the dietary methionine requirements of red drum (*Sciaenops ocellatus*) [34] and stinging catfish (*Heteropneustes fossilis*) [35]. Therefore, the relatively high cysteine content in our study compared to that reported by Alam et al. [30] lowered the dietary methionine requirement of olive flounder in this study.

Supplementing FM with SPC increased ∑SFA but decreased ∑MUFAs and ∑n-3 HUFAs. The levels of n-3 HUFAs, including EPA and DHA, play critical roles in the growth and health of fish [36,37]. The dietary ∑n-3 HUFA (8.16–10.20% of the total FA) requirement of olive flounder [31] was met in the Con and SPC_25_ diets but not the SPC_50_ diet. Nevertheless, lower ∑n-3 HUFA content in the SPC_50_ diet was not detrimental to growth performance. In a previous study, when FM was replaced with up to 25% (7.31% of total fatty acids) SPC, there was a difference in ∑n-3 HUFA content in the Con (11.4% of total fatty acids) but no difference in growth performance [38]. Similarly, Li et al. [39] reported a difference in ∑n-3 HUFA content (Con, 15.39%; SPC25, 11.30% of the total fatty acids) when up to 25% of FM was replaced with tuna or chicken byproduct with no differences in growth performance. Different levels of ∑n-3 HUFA content in the diet (0.26% diet, 6.12%; 0.52% diet, 12% of total fatty acids) did not change the growth performance of juvenile grass carp (*Ctenopharyngodon idellus*) [40]. Furthermore, excessive dietary ∑n-3 HUFA content does not negatively affect the growth of juvenile flounder [40,41].

DFI, FE, PER, and PR were not affected by substituting SPC for FM in this study, suggesting that substituting up to 50% of the FM with SPC in diets supplemented with limited AAs does not decrease DFI or feed utilization (FE, PER, and PR) in this species. Previous studies [15,42] have reported that when 30% of the FM is replaced with SPC and the diet is supplemented with limited AAs, there are no differences in feed availability of olive flounder. In another study, no difference in feed utilization was observed when up to 40% of the FM was replaced with SPC in the yellow croaker diet [11]. In Kokou et al. [19], no difference in feed utilization of gilthead sea bream was observed when up to 60% of the FM was replaced with SPC in the diet. Similarly, in Li et al. [43], no difference in feed utilization was reported when up to 60% of the FM was replaced with SPC in the starry flounder (*Platichthys stellatus*) diet. 

Biological indices (CF, VSI, and HSI) are commonly used to evaluate well-being, obesity, and health of fish. The VSI and HSI indicate the dietary nutritional utilization of fish and their health condition [44]. The biological indices of olive flounder remained unchanged after replacing up to 50% of the FM with SPC in a diet supplemented with limited AAs in this study, indicating that SPC can replace FM without adversely affecting the biological indices of this species. Similarly, substituting a plant protein source for FM does not affect the biological indices of fish [45,46].

Moisture and crude protein of the dorsal muscle were not affected by replacing FM with SPC in this experiment. However, higher crude lipid and lower ash content were found fish fed the SPC_50_ diet compared to the other groups, probably resulting from the lower levels of these components in that food preparation. Many studies have reported on the body composition, AAs, and FAs of fish fed experimental diets [39,41,43,47,48]. However, there are conflicting results on the effect of replacing FM with a plant protein source [49]. Gene expression varies according to the type, amount, and quality of the protein in the diet [50,51]. IGF plays a crucial role in regulating growth, development, and nutritional metabolism in teleost fish [50,52]. Expression of the IGF-I gene in the liver of olive flounder fed combined plant (wheat gluten and SPC) and animal (tankage meal and poultry by-product meal) protein for 20 weeks was comparable to fish fed a 65% FM-based diet, which reflected the growth of fish, but increasing the FM replacement level in the diet decreased the expression of the IGF-I gene in the liver [52]. The findings from the present study at the end of the 140-day feeding trial, expression of growth (IGF and GFB-3) genes in the brain of fish fed the Con diet was higher than fish fed the other diets. Although the growth of fish fed the experimental diets was similar, we assumed that fish fed the Con diet would achieve higher growth than those fed the other diets as the rearing period was extended.

IL-8 presents antigens and stimulates neutrophil chemotaxis [53]. It is detectable when the immune system forms in olive flounder [54]. Caspases play an important role in the immune response and apoptosis and also maintain cell homeostasis [55,56]. A previous study [45] reported that the immune-related gene expression levels in olive flounder were not significantly different when fish were fed a diet containing different levels of FM. In this study, immune (IL-8 and caspase) gene expression in the brains of fish fed the Con diet was higher than in fish fed the other diets, indicating that immune activity might not be properly achieved, and that diet affected the activation and apoptosis of inflammatory cells of fish when fish are fed a low FM-based diet.

Furthermore, Jia et al. [57] reported that excessive or prolonged stress causes physiological disorders, immunosuppression, and reduced growth. Heat shock protein 70 (HSP70) maintains homeostasis in response to various stressors [58] and aids the recovery of damaged cells and proteins [59]. The SOD, GPX, PRX, and TRX genes, which are involved in antioxidant responses, were used to measure antioxidant-related stress. All of these genes, except HSP70 in the brain of fish fed the Con diet, were higher than those of fish fed the SPC_25_ and SPC_50_ diets. HSP70 increased to the corresponding level in a previous study on black sea bream (*Acanthopagrus schlegelii*) but then decreased after HUFA supplementation [60]. Ji et al. [40] reported that excessive HUFA supplementation may be related to oxidative stress in grass carp. This may explain the results of this study so further research is needed. 

Digestibility is an essential factor in the growth of fish. The expression of digestive enzyme genes (Amy, chymo-TRY2, TRY2, TRY3, and lipase) in the stomach and middle intestine of olive flounder was not different in response to the dietary treatments. However, the expression of Amy in the middle intestine decreased as the amount of FM replaced with SPC increased because carbohydrates are normally digested in the intestine [4]. The Chymo-TR2, TRY2, and TRY3 genes and lipase are involved in protein and lipid digestion, respectively. The expressions of both Chymo-TRY2 and trypsin depend on the quality of feed and increase in carnivorous marine fish with high protein availability [4,61]. The expressions of chymo-TRY2, TRY2, and TRY3 genes in the stomach of olive flounder were higher in fish fed the SPC_25_ diet, and the expression of the lipase gene in the stomach of fish fed the SPC_25_ diet was similar to that of fish fed the Con and SPC_50_ diets. The highest chymo-TRY2 expression was observed in the middle intestine of fish fed the Con and SPC_25_ diets, and the expression of TRY2 and lipase was highest in those fed the SPC_25_ diet. No significant difference in the growth performance of olive flounder was observed but it tended to be slightly higher in fish fed the SPC_25_ diet because of the higher expression of digestive-related genes in the stomach and middle intestine. According to previous study [62] reported that lipase in the intestine of fish fed a diet in which FM was replaced with soybean meal was lower than that of fish fed a 42% FM-based diet, indicating the effect of high soybean meal content in the diet on impaired bile secretion. In this study, the lipase activity of fish fed the SPC_50_ diet was lower than that of fish fed the SPC_25_ diet, which was presumed to be affected by SPC. 

## 5. Conclusions

Up to 50% of the FM was substituted with SPC in diets supplemented with AAs without adversely deteriorating growth performance, feed availability, or the biological indices of olive flounder. However, substituting 25% FM with SPC was most effective based on the expression of growth, immune, stress, and digestive enzyme-related genes. Therefore, based on a comprehensive consideration of experimental results and economic values, it is considered effective to replace approximately 25% of the FM content in the diet.

## Figures and Tables

**Figure 1 animals-14-03039-f001:**
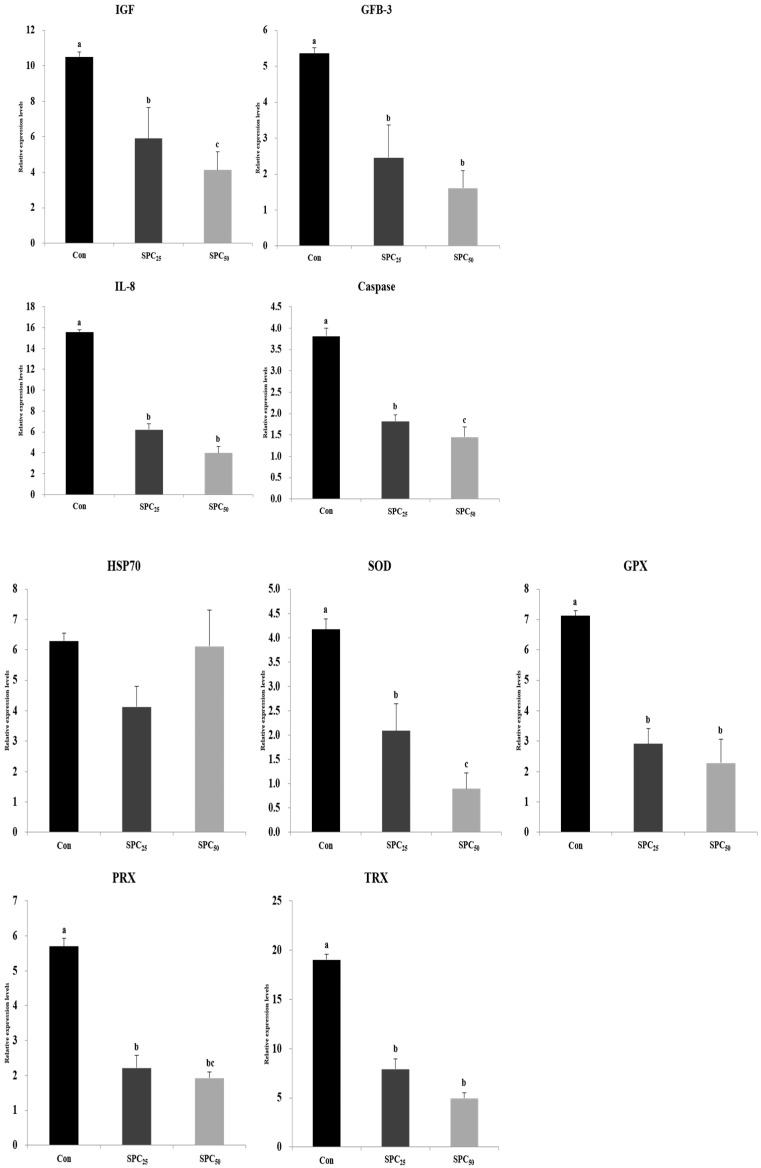
Expression levels of growth, immune, stress, and antioxidant stress-related genes in the brain of olive flounder. The initial (M0-C) value for each gene was set to 1.0. Values (mean of triplicates ± SE) in the same column sharing the same superscript letter are not significantly different (ANOVA, *p* < 0.05). (IGF, *p* = 0.001; GFB-3, *p* = 0.001; IL-8, *p* = 0.001; caspase, *p* = 0.001; HSP70, *p* = 0.05; SOD, *p* = 0.001; GPX, *p* = 0.05; PRX, *p* = 0.001; TRX, *p* = 0.001). IGF, insulin-like growth factor; GFB-3, growth factor beta-3-like protein; IL-8, interleukin-8; HSP70, heat shock protein 70; SOD, superoxide dismutase; GPX, glutathione peroxidase; PRX, peroxiredoxin; TRX, thioredoxin.

**Figure 2 animals-14-03039-f002:**
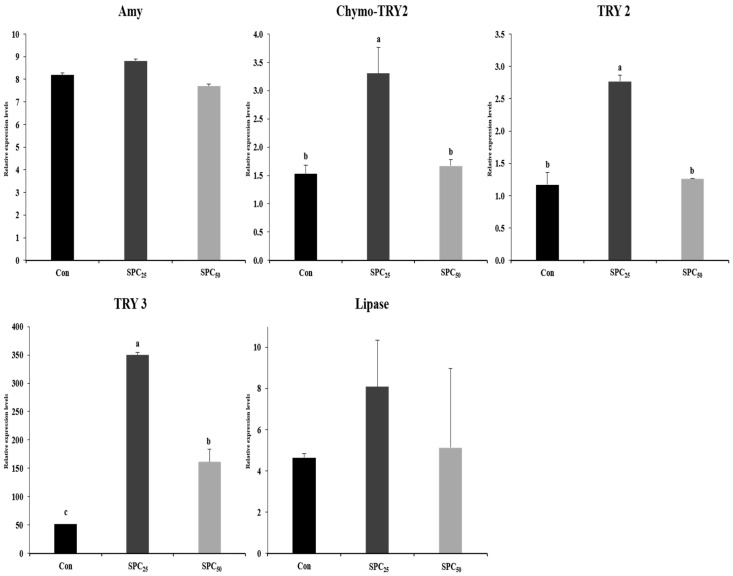
Expression levels of the digestive enzyme genes in the stomach of olive flounder. The initial (M0-C) value for each gene was set to 1.0. Values (mean of triplicates ± SE) in the same column sharing the same superscript letter are not significantly different (ANOVA, *p* < 0.05). (Amy, *p* = 0.05; Chymo-TRY2, *p* = 0.02; TRY2, *p* = 0.02; TRY3, *p* = 0.001; lipase, *p* = 0.05). Amy, α-amylase; chymo-TRY2, chymo-trypsinogen 2; TRY2, trypsinogen 2; TRY3, trypsinogen 3.

**Figure 3 animals-14-03039-f003:**
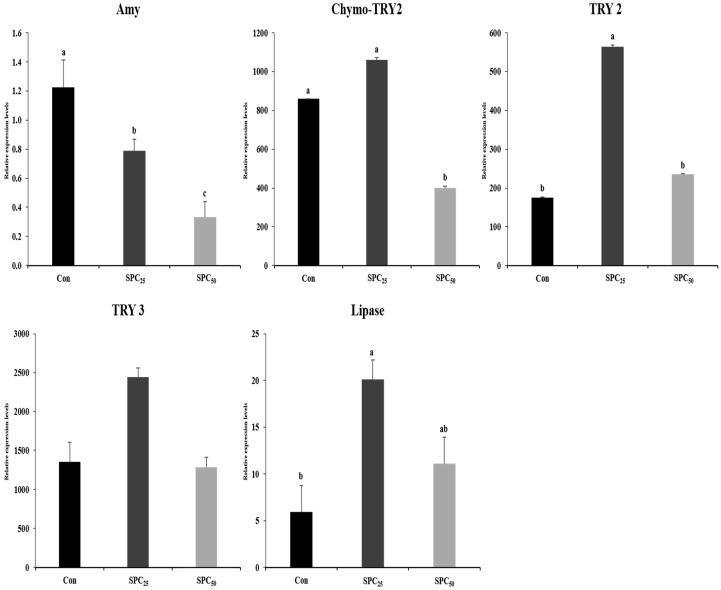
Expression levels of digestive enzyme genes in the middle intestine of olive flounder. The initial (M0-C) value for each gene was set to 1.0. Values (mean of triplicates ± SE) in the same column sharing the same superscript letter are not significantly different (ANOVA, *p <* 0.05). (Amy, *p* = 0.001; Chymo-TRY2, *p* = 0.05; TRY2, *p* = 0.001; TRY3, *p* = 0.05; lipase, *p* = 0.02). Amy, α-amylase; chymo-TRY2, chymo-trypsinogen 2; TRY2, trypsinogen 2; TRY3, trypsinogen 3.

**Table 1 animals-14-03039-t001:** Ingredients and proximate composition of the experimental diets (%, DM basis).

	Experimental Diet		
	Con	SPC_25_	SPC_50_
Ingredient (%, DM)			
Low-temperature fishmeal (LT FM) ^a^	60.0	45.0	30.0
Soy protein concentrate (SPC) ^b^		9.0	18.0
Wheat gluten	9.5	14.25	19.0
Wheat flour	15.5	15.5	15.5
Starch	5.6	3.93	2.26
Fish oil	4.9	6.25	7.6
Mono-calcium phosphate (MCP) ^c^		1.05	2.1
Vitamin C	0.5	0.5	0.5
Mineral premix ^d^	2.0	2.0	2.0
Vitamin premix ^e^	1.0	1.0	1.0
Betaine	0.5	0.5	0.5
Choline	0.5	0.5	0.5
Lysine (99%) ^f^		0.375	0.75
Methionine (99%) ^g^		0.145	0.29
Nutrients (%, DM)			
Dry matter	96.3	96.8	96.3
Crude protein	53.9	53.1	52.8
Crude lipid	9.3	9.5	10.6
Ash	13.3	12.7	10.8
Carbohydrate ^h^	23.5	24.7	25.8
Gross energy (GE) (kcal/g) ^i^	4.800	4.845	4.992

^a^ Low-temperature fishmeal (LT FM) (crude protein: 72.0%, crude lipid: 12.0%, ash: 11.0%) was purchased from FF Skagen (Havnevagtvej, Skagen, Denmark) [USD 1.75/kg FM; 1 USD = 1300 KRW]. ^b^ Soy protein concentrate (SPC) (crude protein: 67.2%, crude lipid: 1.0%, ash: 6.3%) was purchased from ADM^®^ (Quincy, IL, USA) (USD 1.05/kg SPC). ^c^ MCP was purchased from Kunmin Chuan Jin Nuo Chemical Co., Ltd. (Kunming, Yunnan, China). ^d^ Mineral premix (g/kg mix): NaCl, 43.3; MgSO_4_·7H_2_, 136.5; NaH_2_PO_4_·2H_2_O, 86.9; KH_2_PO_4_, 239; CaHPO_4_, 135.3; Ferric citrate, 29.6; ZnSO_4_·7H_2_O, 21.9; Ca-lactate, 304; CuCl, 0.2; AlCl_3_·6H_2_O, 0.15; KI, 0.15; MnSO_4_·H_2_ O, 2.0; CoCl_2_·6H_2_O, 1.0. ^e^ Vitamin premix (g/kg mix): L-ascorbic acid, 121.2; DL-α-tocopherol acetate, 18.8; thiamine hydrochloride, 2.7; riboflavin, 9.1; pyridoxine hydrochloride, 1.8; niacin, 36.4; Ca-D-pantothenate, 12.7; myo-inositol, 181.8; D-biotin, 0.27; folic acid, 0.68; p-aminobenzoic acid, 18.2; menadione, 1.8; retinyl acetate, 0.73; cholecalciferol, 0.003. ^f^ Lysine was purchased from MPbio (Irvine, CA, USA). ^g^ Methionine was purchased from MPbio. ^h^ Carbohydrate was calculated by the difference [100 − (crude protein + crude lipid + ash)]. ^i^ Gross energy (GE) (kcal/g) was calculated by the difference [100 − (crude protein + crude lipid + ash)] and 4 kcal/g for crude protein and carbohydrate, and 9 kcal/g for lipid [24].

**Table 2 animals-14-03039-t002:** Oligonucleotide primers used for quantitative real-time PCR.

Target	Sequence (5′-3′)	Gen Bank
IGF	F.P = 5′-CGGCGCCTGGAGATGTACTG-3′	AF016922.2
	R.P = 5′-TGTCCTACGCTCTGTGCCCT-3′	
GFB-3	F.P = 5′-CTCAAGACCTGGAACCTCTCACTAT-3′	KF723424.1
	R.P = 5′-CTCAGCTACACTTGCAAGACTTGAC-3′	
IL-8	F.P = 5′-GTTGTTGCTGTGATGGTGCT-3′	AB809047.1
	R.P = 5′-GCCGGTATCTTTCAGAGTGG-3′	
Caspase	F.P = 5′-GCACATGGACATCCTGAGTG-3′	AB247499.1
	R.P = 5′-AGGCTGCTCATTTCACTGCT-3′	
HSP70	F.P = 5′-TCCTCATGGGTGACACTTCG-3′	AB010871.1
	R.P = 5′-TTGTCCTTGGTCATGGCTCT-3′	
SOD	F.P = 5′-GGGAATGTCACTGCTGGAAAA-3′	EF681883.1
	R.P = 5′-CCAATAACTCCACAGGCCAGAC-3′	
GPX	F.P = 5′-GAAGGTGGATGTGAATGGGAAG-3′	EU095498.1
	R.P = 5′-TCTGCCTCGATATCAATGGTAAGG-3′	
PRX	F.P = 5′-TCTCCTACAGCAAACAGCAC-3′	DQ009987.1
	R.P = 5′-CCAGGAAGTGACACCATCAA-3′	
TRX	F.P = 5′-TGGACAGAGGCGAGGCTACT-3′	XM020095833.1
	R.P = 5′-ACCCAAAGACCAAACCACACAC-3′	
Amy	F.P = 5′-CACTCTTCATGTGGAAGCTGGTTC-3′	KJ908179
	R.P = 5′-CCATAGTTCTCAATGTTGCCACTGC-3′	
Chymo-TRY2	F.P = 5′-ACTACACCGGCTTCCACTTC-3′	AB029754
	R.P = 5′-GAACACCTTGCCAACCTTCATG-3′	
TRY2	F.P = 5′-ATCGTCGGAGGGTATGAGTG-3′	AB029751
	R.P = 5′-CATCCAGAGACTGTGCACATG-3′	
TRY3	F.P = 5′-TATGAGTGCACGCCCTACTC-3′	AB029752
	R.P = 5′-GTTCTCACAGTCCCTCTCAGAC-3′	
Lipase	F.P = 5′-ATGGGAGAAGAAAATATCTTATTTTTGA-3′	HQ850701
	R.P = 5′-TACCGTCCAGCCATGTATCAC-3′	

Insulin-like growth factor, IGF; growth factor beta-3-like protein, GFB-3; interleukin-8, IL-8; caspase; heat shock protein 70, HSP70; superoxide dismutase, SOD; glutathione peroxidase, GPX; peroxiredoxin, PRX; thioredoxin, TRX; α-Amylase, Amy; chymo-trypsinogen 2, chymo-TRY2; trypsinogen 2, TRY2; trypsinogen 3, TRY3.

**Table 3 animals-14-03039-t003:** Amino acid (g/kg of the diet) profiles of the experimental diets.

	Ingredients		Requirement	Experimental Diet		
	FM	SPC		Con	SPC_25_	SPC_50_
Essential amino acids (EAA) (g/kg)						
Arginine	36.6	45.5	20.4–21.0 ^1^	29.4	27.2	26.3
Histidine	13.4	16.4		9.6	11.5	12.1
Isoleucine	28.9	31.1		22.3	21.6	21.0
Leucine	47.4	50.3		39.4	37.5	36.3
Lysine	50.7	41.3	1.50–21.0 ^2^	36.3	33.6	32.4
Methionine	17.7	6.7	14.4–14.9 ^3^	13.7	12.3	11.7
Phenylalanine	25.3	32.1		22.6	22.8	23.2
Threonine	25.8	24.9		21.7	19.2	17.7
Valine	33.8	31.9		26.7	24.5	23.5
∑EAA ^4^	279.6	280.2		221.7	210.2	204.2
Non-essential amino acids (NEAA) (g/kg)						
Alanine	40.1	27.1		31.7	26.6	23.0
Aspartic acid	59.0	72.6		44.8	40.8	38.5
Cysteine	3.5	5.9	0.6 ^3^	6.1	3.0	2.8
Glutamic acid	89.6	127.8		97.6	111.4	119.3
Glycine	42.9	27.6		33.5	28.2	24.4
Proline	21.8	33.1		30.9	35.5	39.3
Serine	22.6	31.7		22.6	22.1	21.9
Tyrosine	17.7	19.3		15.6	14.8	14.6
∑NEAA ^5^	331.5	384.5		282.8	282.4	283.8

Con: 60% fishmeal-based diet; LF1: dietary replacement of 25% fishmeal with soy protein concentrate; LF2: dietary replacement of 50% fishmeal with soy protein concentrate. ^1,2,3^ Data were obtained from Alam et al. [28], Forster and Ogata [29], and Alam et al. [30], respectively. ^4^ ∑EAA: Total essential amino acid contents. ^5^ ∑NEAA: Total non-essential amino acid contents.

**Table 4 animals-14-03039-t004:** Fatty acid (g/kg of total fatty acids) profiles of the experimental diets.

	Ingredient		Requirement	Experimental Diet		
	FM	SPC		Con	SPC_25_	SPC_50_
C14:0	59.3			34.8	31.9	26.2
C16:0	241.9			205.1	202.9	187.1
C18:0	49.1			197.7	222.6	247.6
C20:0	15.3					
C22:0	19.4					
∑SFA ^1^	384.9			437.6	457.4	460.9
C17:1n-7				48.6	46.5	42.9
C18:1n-9				28.5	28.0	27.0
C24:1n-9	15.2					
∑MUFA ^2^	15.2			77.1	74.5	69.9
C18:2n-6	118.6			178.3	210.3	260.4
C18:3n-6				29.3	31.3	35.2
C20:2n-6	37.9					
C20:3n-6				18.1		
C20:4n-6	44.2					
C20:5n-3	120.6					
C22:2n-6	0.9			68.4	65.0	49.7
C22:6n-3	169.5			117.8	108.3	77.8
∑n-3 HUFA ^3^	491.8		8.16–10.20 ^4^	117.8	108.3	77.8
Unknown	139.7			73.4	53.2	46.1

Con: 60% fishmeal-based diet; LF1: dietary replacement of 25% fishmeal with soy protein concentrate; LF2: dietary replacement of 50% fishmeal with soy protein concentrate. ^1^ ∑SFA: Total saturated fatty acid contents. ^2^ ∑MUFA: Total monounsaturated fatty acid contents. ^3^ ∑n-3 HUFA: Total n-3 highly unsaturated fatty acid contents. ^4^ Data were obtained from Kim and Lee [30].

**Table 5 animals-14-03039-t005:** Survival (%), weight gain (g/fish), and specific growth rate (SGR) of olive flounder fed the experimental diets for 140 days.

Experimental Diet	Initial Weight (g/Fish)	Final Weight (g/Fish)	Survival (%)	Weight Gain (g/Fish)	SGR (%/Day)
Con	721.4 ± 6.95	1112.5 ± 28.80	96.3 ± 1.79	390.8 ± 24.95	0.31 ± 0.016
SPC_25_	729.4 ± 5.63	1170.3 ± 90.10	94.7 ± 4.09	442.3 ± 86.24	0.34 ± 0.048
SPC_50_	732.5 ± 3.61	1131.9 ± 24.47	96.3 ± 2.49	400.6 ± 24.29	0.31 ± 0.015
*p*-value	*p* = 0.9	*p* = 0.9	*p* = 0.05	*p* = 0.9	*p* = 0.05

Values (mean of triplicates ± SE) in the same column sharing the same superscript letter are not significantly different (ANOVA, *p* < 0.05).

**Table 6 animals-14-03039-t006:** Daily feed intake (DFI), feed efficiency (FE), protein efficiency ratio (PER), protein retention (PR), condition factor (CF), viscerosomatic index (VSI), and hepatosomatic index (HSI) of olive flounder fed the experimental diets for 140 days.

Experimental Diet	DFI (%/Day)	FE	PER	PR (%)	CF (g/cm^3^)	VSI (%)	HSI (%)
Con	0.48 ± 0.018	0.63 ± 0.024	1.16 ± 0.045	37.36 ± 1.510	1.04 ± 0.247	3.99 ± 0.574	1.47 ± 0.282
SPC_25_	0.46 ± 0.004	0.70 ± 0.074	1.32 ± 0.139	42.62 ± 3.703	1.16 ± 0.283	3.59 ± 0.263	1.57 ± 0.344
SPC_50_	0.44 ± 0.009	0.69 ± 0.021	1.30 ± 0.039	42.90 ± 1.256	1.04 ± 0.083	3.45 ± 0.356	1.44 ± 0.150
*p*-value	*p* = 0.05	*p* = 0.05	*p* = 0.5	*p* = 0.9	*p* = 0.6	*p =* 0.8	*p* = 0.9

Values (mean of triplicates ± SE) in the same column sharing the same superscript letter are not significantly different (ANOVA, *p* < 0.05).

**Table 7 animals-14-03039-t007:** Proximate composition (%, wet weight) of the dorsal muscle of olive flounder fed the experimental diets for 140 days.

Experimental Diet	Moisture	Crude Protein	Crude Lipid	Ash
Con	72.87 ± 0.020	22.44 ± 0.240	2.37 ± 0.400 ^b^	1.57 ± 0.005 ^a^
SPC_25_	72.69 ± 0.330	22.26 ± 0.500	2.61 ± 0.025 ^b^	1.55 ± 0.025 ^a^
SPC_50_	71.95 ± 0.085	21.91 ± 0.045	4.44 ± 0.145 ^a^	1.50 ± 0.005 ^b^
*p*-value	*p* = 0.1	*p* = 0.4	*p* = 0.04	*p* = 0.05

Values (mean of triplicates ± SE) in the same column sharing the same superscript letter are not significantly different (ANOVA, *p <* 0.05).

## Data Availability

Data are contained within the article. The original contributions presented in the study are included in the article, and further inquiries can be directed to the corresponding author.

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
