# Peer review of "Effects of Substituting Fishmeal with Soy Protein Concentrate Supplemented with Essential Amino Acids in the Olive Flounder (Paralichthys olivaceus) Diet on the Expression of Genes Related to Growth, Stress, Immunity, and Digestive Enzyme"

_animals, 2024, doi:10.3390/ani14203039_

Round 1

Reviewer 1 Report

Comments and Suggestions for Authors

Dear Authors

The ms entitled " Effects of Substituting Fishmeal with Soy Protein Concentrate

Supplemented with Essential Amino Acids in the Olive Flounder (Paralichthys olivaceus) Diet on the Expression of Growth, Stress, Immune, and Digestive Enzyme-Related Genes” is an interesting, well-done, and carefully research paper that provides information along low fishmeal diet and growth/physiology of olive flounder,

The results presented in this study are interesting and well-structured, and they provide some interesting initial results,  along with a low fishmeal diet and growth -physiology of olive flounder.

In addition to these general points, there are some specific points which are addressed as follows:

Introduction

  • All animal species mentioned for the first time in this article should be written with the author's last name and the year of description. When mentioned again, the generic name should be shortened to one letter.

Material and methods

·       Line 78-79: the authors should write what the oxygen supplied was; it was saturated; you should mention in the text

·       The authors should briefly explain why they chose these replacement rates for the experiment (Lines 83-87)

·       The growth indices (biological indices) listed below the tables (table 5, table 6)  should be removed and reported in paragraph 2.3. The abbreviations of the parameters studied are also given in the tables and calculation equations. In the text, they are indicated by their abbreviations.

Results

·       Tables should be written within the text in the paragraph where the results are interpreted, not in the paragraph heading, such as  in paragraph 3.2, “Survival and growth performance (table 5).” The authors can correct it throughout the text in all the results.

·       It would be helpful to present the statistical analysis in a better way in the tables in order to clarify the statistical difference between the treatments (tables 1-7)

·       The authors above should do the same with figures and refer to them in the text and not in the headings throughout the ms. (example paragraph 3.5 Expression Analysis…. (Fig.1) etc.

·       In the text, for example, in the last line of table 5, or P>0.9 etc. This is not correct. It is necessary to write the actual p-value for the example p=0.4>0.9). Furthermore, the authors under Table 5 for example, write “(p<0.05)” without reporting the statistical test that they have done, such as (ANOVA, p>0.05)

Discussion

·       Line 306, In a previous study ……please rewrite as …Previous studies [37]…..

·       Line 388, Kim et al. and Park et al. .. please rewrite as According to previous studies [15,41] were reported …….

·       Line 345, In our study,……please rephrase as The findings from the present study at the end……

·       Line  352, A Previous study, erase the “A” and rewrite as Previous study [44]

·       Line 359 Please add. Furthermore, Jia et al. [56]

·       Line 384, lease ad According to ….. and rephrase the sentence  

Conclusion:

·       It would help if you showed how the research questions have been thoroughly examined and explained.

·       The research questions are thoroughly examined and explained in the discussion section.

·       Consider adding the theoretical and practical implications of the study's findings.

Author Response

Reviewer 1
First of all, thank you for taking the time to review this manuscript.
Your feedback has helped us to improve the paper and develop it into a more effective and polished piece of work.

Introduction

All animal species mentioned for the first time in this article should be written with the author's last name and the year of description. When mentioned again, the generic name should be shortened to one letter.
- I accept your opinion and modified it.
Material and methods

·Line 78-79: the authors should write what the oxygen supplied was; it was saturated; you should mention in the text 
- I accept your opinion and modified it to ‘dissolved oxygen (8.5±1.0 mg/L; mean±SD)’

·The authors should briefly explain why they chose these replacement rates for the experiment (Lines 83-87)
 – It was set by referring to the replacement rate of the previous papers and marking it.

·The growth indices (biological indices) listed below the tables (table 5, table 6) should be removed and reported in paragraph 2.3. The abbreviations of the parameters studied are also given in the tables and calculation equations. In the text, they are indicated by their abbreviations.
 - I accept your opinion and modified it.

Results

·Tables should be written within the text in the paragraph where the results are interpreted, not in the paragraph heading, such as in paragraph 3.2, “Survival and growth performance (table 5).” The authors can correct it throughout the text in all the results.
- I accept your opinion and modified it.

·It would be helpful to present the statistical analysis in a better way in the tables in order to clarify the statistical difference between the treatments (tables 1-7)
- I accept your opinion and modified it.

·The authors above should do the same with figures and refer to them in the text and not in the headings throughout the ms. (example paragraph 3.5 Expression Analysis…. (Fig.1) etc.
 - I accept your opinion and modified it.

·In the text, for example, in the last line of table 5, or P>0.9 etc. This is not correct. It is necessary to write the actual p-value for the example p=0.4>0.9). Furthermore, the authors under Table 5 for example, write “(p<0.05)” without reporting the statistical test that they have done, such as (ANOVA, p>0.05)
I accept your opinion and modified it to P value in tables ex) P=0.9 and ‘Values (mean of triplicates±SE) in the same column sharing the same superscript letter are not significantly different (ANOVA, P< 0.05)’.

Discussion

·Line 306, In a previous study ……please rewrite as …Previous studies [37]….. 
- I accept your opinion and modified it.

·Line 388, Kim et al. and Park et al. .. please rewrite as According to previous studies [15,41] were reported ……. 
- I accept your opinion and modified it

·Line 345, In our study,…… please rephrase as The findings from the present study at the end……
- I accept your opinion and modified it

·Line 352, A Previous study, erase the “A” and rewrite as Previous study [44] 
- I accept your opinion and modified it

·Line 359 Please add. Furthermore, Jia et al. [56]
- I accept your opinion and modified it

·Line 384, Please ad According to ….. and rephrase the sentence 
- I accept your opinion and modified it

Conclusion:

·It would help if you showed how the research questions have been thoroughly examined and explained.
·The research questions are thoroughly examined and explained in the discussion section.
·Consider adding the theoretical and practical implications of the study's findings.
- I accept your opinion and modified it

Reviewer 2 Report

Comments and Suggestions for Authors

General comments: this is a competent paper providing interesting data on the substitution of fishmeal with Soya proteins. The authors have carried out all the control determinations of the diet and composition of the fish under the three different regimes. A few general points require attention especially in the methodology.

Specific comments

Title: The title of the paper is too long and really does not indicate  to the author the true essence of the paper. Growth, Stress and immune are not an expression. Only genes can be expressed. The running title also needs changing.

Page 1 line 10: please edit the abstract to suggest that first of all biological parameters have been measured and then the corresponding gene expression

Page 2 line 68: please state the total number of fish used in the study. Are you able to measure the fish to the second decimal place? Please round up the numbers. In total I assume if nine tanks are used with 100 fish in each tank then 900 fish were used. Is this correct?

Page 2 line 78: Please explain why the temperature range was so large and did this affect the feeding of the fish?

Page 3 line 115:Please state which biological indices you mean. Please make it clear that the fish were weighed at the beginning and end of the experiment.

Page 7 table 5: please define what the mean of triplicate's describes. Does this mean that the 900 fish themselves were not measured only a small sample?

Page 8 table 6: Again please indicate what triplicates means

Page 9 figure 1:The legend on the Y axis is far too small. Please state in the figure legend how the relative expression level was calculated

Page 10 and 11Figures 2 and 3 : same comment as above

Author Response

Reviewer 2
First of all, thank you for taking the time to review this manuscript.
Your feedback has helped us to improve the paper and develop it into a more effective and polished piece of work.

Specific comments

Title: The title of the paper is too long and really does not indicate to the author the true essence of the paper. Growth, Stress and immune are not an expression. Only genes can be expressed. The running title also needs changing.
 - Title change to 'Effects of Substituting Fishmeal with Soy Protein Concentrate Supplemented with Essential Amino Acids in the Olive Flounder (Paralichthys olivaceus) Diet on the Expression of Genes related to Growth, Stress, Immunity, and Digestive Enzyme'.
 - Running title changed to 'Growth/physiological changes with low fishmeal diet supply of olive flounder'.

Page 1 line 10: please edit the abstract to suggest that first of all biological parameters have been measured and then the corresponding gene expression
 - I attempted to make the requisite modifications, but was unable to do so due to the constraints imposed by the limited number of characters.

Page 2 line 68: please state the total number of fish used in the study. Are you able to measure the fish to the second decimal place? Please round up the numbers. In total I assume if nine tanks are used with 100 fish in each tank then 900 fish were used. Is this correct?
 - total 900 fish was added in the article; and we didn't measure the fish to the decimal place, but it is presumed that this was misunderstood, as the lack of a mark for 'initial mean weight' may have led to confusion.

Page 2 line 78: Please explain why the temperature range was so large and did this affect the feeding of the fish?
 -  The substantial alteration in water temperature can be attributed to the fact that the experiment was conducted for approximately four months using natural seawater.

Page 3 line 115: Please state which biological indices you mean. Please make it clear that the fish were weighed at the beginning and end of the experiment.
 - I accept your opinion and modified it
Page 7 table 5: please define what the mean of triplicate's describes. Does this mean that the 900 fish themselves were not measured only a small sample?
 - The whole fish was measured through this sentence 'All live fish from each tank were counted and weighed collectively to measure survival and weight gain at the beginning and end of the experiment.' and the use of three replicates per experimental group means triplicate's. So I have updated the text in the ‘2.1 Experimental Conditions’ section to read ‘(n = 3 tanks per treatment)’.

Page 8 table 6: Again please indicate what triplicates means
- I accept your feedback and have updated the text in the ‘2.1 Experimental Conditions’ section to read ‘(n = 3 tanks per treatment)’.

Page 9 figure 1:The legend on the Y axis is far too small. Please state in the figure legend how the relative expression level was calculated
- The relative expression level is indicated on the Y-axis as a legend based on M0-C. Furthermore, each table clearly states that tha data is based on M0-C, with ‘The initial (M0-C) value for each gene was set to 1.0’.
Page 10 and 11 Figures 2 and 3: same comment as above
- The relative expression level is indicated on the Y-axis as a legend based on M0-C. Furthermore, each table clearly states that tha data is based on M0-C, with ‘The initial (M0-C) value for each gene was set to 1.0’.
